META-RESEARCH ARTICLE

# Identifying barriers and enablers to rigorous conduct and reporting of preclinical laboratory studies

**Manoj M. Lalu** [1,2,3]*, **Justin Presseau**[1,4,5], **Madison K. Foster**[1], **Victoria T. Hunniford**[1], **Kelly D. Cobey**[4,6], **Jamie C. Brehaut**[1,4], **Carolina Ilkow**[7,8], **Joshua Montroy**[1], **Analyssa Cardenas**[1], **Ayni Sharif**[1], **Matthew S. Jeffers**[1], **Dean A. Fergusson**[1,4]

1 Clinical Epidemiology Program, The Ottawa Hospital Research Institute; Ottawa, Ontario, Canada,
2 Regenerative Medicine Program, The Ottawa Hospital Research Institute; Ottawa, Ontario, Canada,
3 Department of Anesthesiology and Pain Medicine, The Ottawa Hospital; Ottawa, Ontario, Canada,
4 School of Epidemiology and Public Health, Faculty of Medicine, University of Ottawa; Ottawa, Ontario, Canada, 5 School of Psychology, University of Ottawa; Ottawa, Ontario, Canada, 6 Meta-Research and Open Science Program, University of Ottawa Heart Institute; Ottawa, Ontario, Canada, 7 Cancer Therapeutics Program, Ottawa Hospital Research Institute; Ottawa, Ontario, Canada, 8 Department of Biochemistry, Microbiology and Immunology, University of Ottawa; Ottawa, Ontario, Canada

* mlalu@toh.ca

**Data Availability Statement:** Data are available in the main text or the supplementary materials. Data and materials pertaining to individual participants will not be shared to protect privacy.

## Abstract

Use of rigorous study design methods and transparent reporting in publications are 2 key strategies proposed to improve the reproducibility of preclinical research. Despite promotion of these practices by funders and journals, assessments suggest uptake is low in preclinical research. Thirty preclinical scientists were interviewed to better understand barriers and enablers to rigorous design and reporting. The interview guide was informed by the Theoretical Domains Framework, which is a framework used to understand determinants of current and desired behavior. Four global themes were identified; 2 reflecting enablers and 2 reflecting barriers. We found that basic scientists are highly motivated to apply the methods of rigorous design and reporting and perceive a number of benefits to their adoption (e.g., improved quality and reliability). However, there was varied awareness of the guidelines and in implementation of these practices. Researchers also noted that these guidelines can result in disadvantages, such as increased sample sizes, expenses, time, and can require several personnel to operationalize. Most researchers expressed additional resources such as personnel and education/training would better enable the application of some methods. Using existing guidance (Behaviour Change Wheel (BCW); Expert Recommendations for Implementing Change (ERIC) project implementation strategies), we mapped and coded our interview findings to identify potential interventions, policies, and implementation strategies to improve routine use of the guidelines by preclinical scientists. These findings will help inform specific strategies that may guide the development of programs and resources to improve experimental design and transparent reporting in preclinical research.

**Funding:** DAF, MML, JCB, JP and KDC were supported by grants from BioCanRx (a Government of Canada funded National Centers of Excellence; Catalyst grant) and the Canadian Vascular Network (a Canadian Institutes of Health Research funded network; Knowledge Translation grant). MML is supported by The Ottawa Hospital Anesthesia Alternate Funds Association and holds a University of Ottawa Junior Research Chair in Innovative Translational Research and the Canadian Anesthesiologist's Society Career Scientist Award. The funders had no role in study design, data collection and analysis, decision to publish, or preparation of the manuscript.

**Competing interests:** The authors have declared that no competing interests exist.

**Abbreviations:** ARRIVE, Animal Research: Reporting of In Vivo Experiments; BCW, Behaviour Change Wheel; DORA, Declaration on Research Assessment; EQIPD, Enhancing Quality In Preclinical Data; ERIC, Expert Recommendations for Implementing Change; NIH, National Institutes of Health; TDF, Theoretical Domains Framework.

## Introduction

Optimal conduct and reporting of preclinical laboratory experiments are critical to support reproducible research [1,2]. Rigorous conduct, which includes key elements such as randomization and blinding, is needed to reduce risk of bias and inflation of effect sizes [2–7]. Similarly, complete reporting of pertinent details is needed so that the rigor of studies can be evaluated and threats to validity appropriately considered. A recent series of articles and editorials describing results from the Reproducibility Project: Cancer Biology [8–11] demonstrate major difficulties in replicating preclinical experiments, with insufficient reporting highlighted as a core challenge. Errington and colleagues observed reduced effect sizes in 112 replication attempts with only 46% of replications demonstrating similar effects as previously published studies [9]. Results of these studies and others have led to calls for improved reporting of preclinical research, to facilitate evaluation and replication, as well as the use of rigorous approaches to address potential research-related bias.

To encourage both rigorous conduct and transparent reporting, the National Institutes of Health (NIH) assembled international stakeholders to reach a consensus on essential reporting items [2] in preclinical animal research. The resulting guidelines were published in 2014 [12] and encompass a core set of 7 items that, at a minimum, should be reported in preclinical publications: (1) use of community-agreed standards in nomenclature and reporting; (2) distinguishing biological and technical replicates; (3) statistics; (4) randomization; (5) blinding; (6) sample size estimation; and (7) inclusion and exclusion criteria. Though labeled as a reporting guideline, as emphasized by the similar Animal Research: Reporting of In Vivo Experiments (ARRIVE) guideline and the recent 2020 ARRIVE 2.0 guideline, such domains represent important considerations for conduct; *"guidelines and associated resources are useful throughout the course of a study"* including *"during study planning"* and *"during the conduct of a study"* [13].

The uptake and impact of such guidelines has been poor [14,15] despite efforts to increase adoption through new evaluations of reproducibility and rigor in grant applications [16], establishment of similar guidelines by other groups [1], and several targeted educational programs [17,18]. The consequences of poor uptake may mean that studies in this field are prone to bias [3–6]. Furthermore, a lack of uptake may mean studies are not being adequately reported, making it difficult for others to evaluate study rigor or attempt replication. Underlying reasons for this poor uptake need to be understood and addressed.

Use of guidelines can be approached as a specific behavior that researchers conducting preclinical experiments could engage in. Framing activities as preclinical research *behaviors* enables us to draw upon decades of understanding on the factors that facilitate and impede behavior—and strategies best suited to address these factors. For example, the "Theoretical Domains Framework" synthesizes theories of behavior change into 14 "domains" [19,20] that reflect major influences on enacting a behavior (TDF; see S1 File for domains and definitions) [21]. Barriers and enablers to behavior identified by the TDF can be combined with other frameworks (Behaviour Change Wheel (BCW); Expert Recommendations for Implementing Change (ERIC)) [21–23] to guide development of strategies to help adopt behaviors of interest [21,24–29]. In the present study, we drew upon this approach to interview preclinical scientists and systematically examine the barriers and enablers to using preclinical guidelines. We also identified strategies that might improve their uptake. Throughout this study, we refer to "using preclinical guidelines" to encompass both implementation of the guidelines during study conduct, as well as reporting [21–23].

## Results

### Participant characteristics

Thirty preclinical researchers participated in an interview. Interviewees were primarily affiliated with our funders (2 federally funded Canadian research networks), thus our final sample

**Table 1. Participant characteristics.**

| Characteristic | Frequency or median (N = 30) |
|---|---|
| **Median age (range) in years** | 42 (27–66) |
| **Gender** | |
| Male | 19 |
| Female | 11 |
| **Career level** | |
| Investigators | 17 |
| Highly qualified personnel* | 13 |
| **Primary area of research** | |
| Cancer | 15 |
| Cardiovascular | 15 |
| **Median experience conducting preclinical in vivo work (range) in years** | 13 (1–42) |
| **Median length of interview (range) in minutes** | 42 (26–69) |

The data underlying this figure can be found in S1 Data.

* Research associates, postdoctoral fellows, PhD students.

included 29 Canadian researchers and 1 American researcher from 13 different institutions. Participant characteristics are found in Table 1.

## Interview findings

Our analysis encompassed 15 TDF domains in total; 14 domains from TDF version 2, and the Nature of Behaviour Domain from TDF version 1. More details are provided in the Methods section. Overall, 12 TDF domains were deemed relevant to implementing preclinical guidelines: knowledge, skills, nature of the behavior, beliefs about consequences, beliefs about capabilities, environmental context and resources, goals, behavioral regulation, reinforcement, social influences, social professional role and identity, and intentions. Among these, issues within 2 domains—*knowledge* and *skills*—were identified primarily as barriers, while issues related to 5 domains—*social influences*, *social professional role and identity*, *behavioral regulation*, *reinforcement*, and *intentions*—were identified primarily as enablers. The remaining 5 domains incorporated both barriers and enablers. Three TDF domains were not identified as relevant: *emotions*; *memory, attention, and decision processes*; and *optimism*.

Four global themes were identified, 2 that reflect barriers and 2 that reflect enablers (themes, sub-themes, relevant TDF domains, specific beliefs, and associated frequency counts are presented in S5 File). These themes are presented alongside illustrative quotes that demonstrate participant beliefs and do not represent the position of the authors on each subject.

**Barrier Theme 1: There is variability in awareness and current practices.** In general, there was varied awareness of preclinical guidelines (e.g., NIH guidelines, ARRIVE [12,30]). Some participants indicated they were aware of some guidelines, while other participants had not heard of any formal guidelines (i.e., for preclinical study conduct and/or reporting). Many participants indicated that they were aware that not all core items were routinely implemented. Some participants shared that they believed that most labs implement these practices, though they may not report them—either because reporting of core practices is not common practice or that lack of reporting is due to word count restrictions for many journals.

*". . .I . . . would challenge you that in many cases the failure to report is not a failure of the authors willingness to report, but an inability of the journal to accept that this information needs to be included in the manuscript."*—Participant 22

The level of formal training on how to reduce bias in a preclinical in vivo study varied across participants. Multiple participants shared that they had not received formal training on preclinical study design, stating that they were mentored, self-taught, or drew from experiences in other research areas (e.g., some had experience in clinical trial design). Those who had received formal training shared that this was obtained through graduate school, workshops, and/or consulting with institutional research services or biostatisticians.

When participants were asked about their role in implementing guideline recommendations on design and reporting, the level of implementation varied across core practices. Specifically, participants were asked about their role in implementing/reporting core practices suggested by the NIH: blinding, randomization, sample size estimation, statistics, inclusion and exclusion criteria, replicates, and in the use of reporting guidelines. Of these practices, most had prior experience with blinding, though some stated that they did not currently use blinding in their research. Most indicated that they (or their team) had experience with randomization of animals into treatment groups, however, when probed further some described pseudo-randomization (e.g., by cage when the experimental unit was the animal). Though arbitrary assignments may seem "random," this is in fact a form of "pseudo" or "simple" randomization that still leaves room for judgment and thus potential for bias (use of a random number generator is recommended by the ARRIVE guidelines as an optimal method for randomization) [31].

*". . .we haven't randomized. I mean the randomization we do is whatever animal is available . . .we don't actually flip a coin and do mutant first, wild type, it's just, it's kind of random in a way that whatever's available we perform the experiment. . . .the labs that I've worked with we've never randomized our samples."*—Participant 19

Likewise, although most stated they had experience with determining or reporting experimental replicates, several suggested that they do not routinely make the distinction between biological and technical replicates. Practices with which participants had the least experience included: a priori determination of inclusion and exclusion criteria, sample size calculation, and the use of reporting guidelines. A priori inclusion and exclusion criteria are important for in vivo experiments; epigenetic and environmental factors can lead to differences in animal populations. Furthermore, a priori inclusion and exclusion criteria can be used to define whether incidental occurrences will be excluded from the analysis (e.g., unrelated illness).

*". . .so we don't necessarily calculate in advance the sample size, doing power analysis, you know this is not something, we probably should do it, I don't see any barrier for doing it except the knowledge barrier. . ."*—Participant 17

*". . .we do have inclusion and exclusion criteria, but they tend to be, post analysis, not, we don't normally decide them, before the start of the experiment because all of the stuff we do is kind of new and sometimes we don't really know what to expect, right, we cross the bridge when we get there. So it's, I would say inclusion/exclusion criteria might be, might be something that we could improve. . ."*—Participant 18

**Barrier Theme 2: Costs and challenges of implementing the guidelines.** Perceived cost and feasibility issues were seen as major barriers to rigorous design. Notably, participants

stated that implementation of the recommended practices would incur financial costs and deplete their often-strained resources. They indicated that it would take them more time to properly implement certain practices as they would require additional work, more planning, and a higher level of organization. Some specifically stated that certain practices (such as blinding, randomization, and sample size estimation) could be challenging and would require additional effort to understand, implement, and ensure that all study personnel (lab technicians and students) were sufficiently trained. Moreover, potential increases in sample sizes would result in greater costs to purchase, house, and care for animals. Participants shared that the time restraints caused by the guideline implementation would either slow down their work—meaning they could not perform as many experiments during a given time period and/or that they would require additional personnel to share the workload or carry out some of these practices—again, requiring additional financial resources.

> "...there needs to be enough flexibility with respect to sample size and replicates... If you can't justify 30 animals to animal care, [if] you can't afford to purchase, house, maintain and then conduct experiments on 30 animals then pay the people who have to do it, then I think it could, in theory, set the institution [of] science up for a failure in that they might miss something that is potentially a building block to a large discovery. It has potential to be prohibitive to a not extremely largely staffed, well-funded lab."—Participant 7

Another barrier was associated with the nature and aim of the preclinical research. Participants stated that for certain study designs, it may not be possible to follow all practices. For example, participants stated that it is not possible to ensure blinding with all animal models or types of interventions as inadvertent unblinding would routinely occur (e.g., use of a food additive in experimental animals, which incidentally dyes tissue, would be readily detectible during harvest procedures). Additionally, participants discussed the nature and aim of exploratory preclinical research (versus confirmatory research), stating that some practices may not be appropriate or applicable to exploratory/discovery preclinical research.

> "If it's something that's at a sort of mature stage...then yes I would say those guidelines are practical and important. For more exploratory experimental studies, where we're just trying to get at a picture of the landscape for everything, we don't really implement those things naturally because we're still trying to sort out technical things or whatnot. And all the data that would come out of it we would still use to guide next steps."—Participant 11

**Enabler Theme 1: Benefits and proficiency increase motivation and intentions.** Another major theme, and an enabler, was that participants believed that there are benefits to implementing increased rigor and transparent reporting in their in vivo research. Many stated that implementing the guidelines would improve robustness, quality, accuracy, and reliability of the data they generate from their research and reduce the possibility of research-related bias. Additionally, participants asserted that by following the guidelines, they hoped to improve the reproducibility of their work. Some suggested this may improve therapeutic development as transparent reporting may optimize evaluation of therapies selected for translation.

> "The most important thing is we want to do good research that is reproducible, rigorous, and with high translational potential for the health of the population."—Participant 2

In general, participants were motivated to implement experimental rigor and transparent reporting if they felt that it was logistically and personally feasible to do so. Indeed, several

suggested these practices were needed to publish—particularly in high-impact and reputable journals. Participants also indicated that requirements by grant agencies and journals could be motivating to themselves and/or to other researchers.

*"I want to do good science. I want what I do to matter in the sense that if I have goals of taking it to a human clinical trial to better the treatment of humans with disease then I've done everything appropriately in a way that I'm sure that what I'm doing is justified to go forward to testing it in humans. That I'm not wasting resources that I'm not putting humans in dangers with the testing of it going down the road. I think that these are all things that are critically important to doing it properly or you can't justify taking it that far."*—Participant 6

When participants were asked to discuss their personal ability to implement core practices, most stated they were very confident or confident in their capabilities to apply all the experimental design methodology. Overall, participants had high intentions to apply the guidelines in their future work or shared that they already implement many of the core practices.

*"Yes, I mean to the best. . .that we can. And I would certainly go back at this and look at it in a little bit more detail to see how similar or different it is to what we're actually doing in practice right now. Just reading some of these things, I can see that there are things we can improve on and maybe work to implement. But some of these other things, biological and technical replicates, those are things we've always included in our experiment design."*—Participant 8

*"Yes all of those have to be followed. We'll be considering every last one of them, just as we have done in the past."*—Participant 21

Almost all participants stated that general education or training would be helpful. Many participants also shared that it would be beneficial for researchers to learn about rigorous design and transparent reporting early in their research career or at the start of training (i.e., as a trainee). When further questioned about the preferred format of a hypothetical information/training session, the most commonly preferred formats included a website or online platform, meeting, or workshops.

*"I think especially for the trainees, there needs to be a certain level of formal training along these lines. Because that doesn't exist now, at all. It's left really up to the PIs to do it, and then the PIs biases get involved in it and those biases are driven by funding and other kinds of factors. And so I think that kind of training has to start really early. The early stages of graduate school and so on. This is the gold standard that we have to subscribe to, and we're expected to follow those rules."*—Participant 6

**Enabler Theme 2: Need for support, resources, and system-level changes.**   Support for personnel and collaborative networks were identified as important factors to implement the guidelines. At the time of the interviews, many participants stated that they conduct their research collaboratively and work in teams—sharing that they involve multiple personnel, laboratory technicians, and consult individuals with different areas of expertise and specializations. When asked who helps and supports them in their research to apply core practices, participants highlighted various individuals who did so: principal investigators, lab managers/supervisors, laboratory staff and technicians, animal care personnel, and students/trainees. Many participants also shared that they consult a biostatistician or that their institution has a biostatical consultation service available.

Multiple participants shared that additional personnel support would enable further implementation. Particularly, participants stated that a designated individual/"champion" to train/teach the core practice methodology, ensure their implementation, and/or provide methodological and statistical support would be valuable. Overall, this theme highlighted the importance of taking a team approach and the involvement of those with expertise, indicating that a lack of such support may hinder implementation of the guidelines.

> *"Everyone needs to be on the same page and. . .it would be good if there was someone who sort of had all this knowledge that you could just go to and get help with for designing your study."*—Participant 10

> *"..so I would say resources, funding, would be one important factor. And then having somebody who can be overseeing it. Then I would say that person ensures that the trainees are aware of this. So that they're implementing it on a day-to-day basis."*—Participant 8

> *"I think really having someone in the lab that's really expert with the guidelines and really know what to do and how to do it would be the best. So then when you arrive in the lab that's doing preclinical research, that person is showing you how to do it so, you learn from then and you learn the best way to do it at the beginning. So, then it becomes routine."*—Participant 15

### Proposed implementation strategies

TDF domains identified from the interviews were mapped to potential strategies to overcome barriers to implementation of the guidelines (Fig 1 and Table 2; further details provided in S6 File).

## Discussion

In this study, we identified barriers and enablers to implementing rigorous experimental design and transparent reporting, from the perspective of preclinical researchers—the intended users of guidelines promoted by the NIH and other agencies. Our work has also allowed us to identify possible strategies for preclinical researchers. Taken together, the findings from this study provide evidence to develop and test behavior-change interventions to improve the uptake of preclinical experimental design and reporting guidelines. These

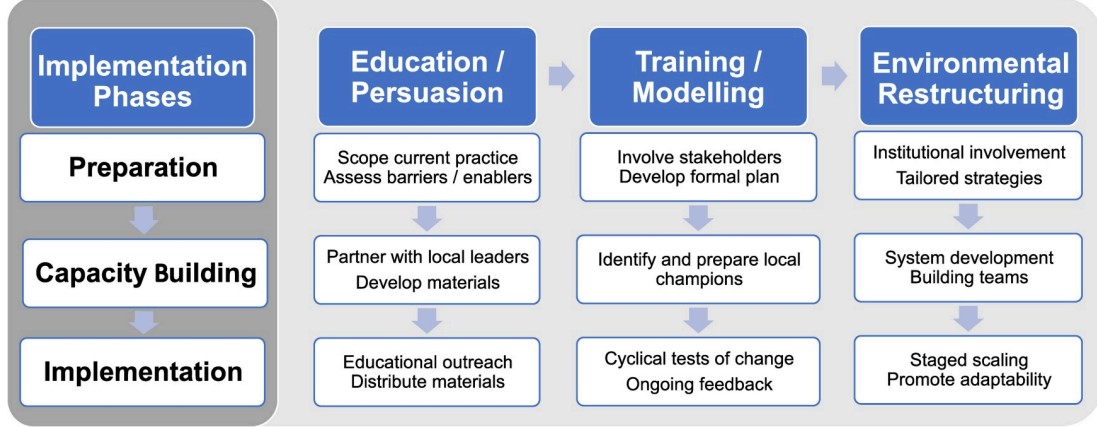

**Fig 1. Proposed phases for implementing rigorous preclinical experimental study design and transparent reporting for preclinical researchers.**

**Table 2. Proposed strategies for preclinical researchers.**

| Implementation phase | Education/persuasion | Training/modeling | Environmental restructuring |
|---|---|---|---|
| **Preparation** | • Discuss with colleagues how they have successfully implemented rigorous experimental design and transparent reporting<br>• Collaborate to identify lab-specific barriers to adoption and potential methods to overcome these barriers<br>• Meet with other labs currently implementing these practices to learn their strategies (e.g., lab tours, shadowing) | • Ask colleagues and stakeholders for input on the implementation plan, provide support and work together to problem solve | • Involve institutional and departmental leaders to help increase scale of implementation<br>• Hold team meetings to assess/evaluate implementation of the guidelines and how members have overcome barriers<br>• Develop a plan with steps to implementing each practice and member roles tailored to the practices that have worked best in other groups |
| **Capacity building** | • Establish collaborative group(s) to study preclinical experimental design and reporting and identify educational needs and resources<br>• Identify funding opportunities that require researchers to prioritize practices outlined by design and reporting guidelines<br>• Collate existing educational resources and literature outlining key aspects that are particularly relevant to your lab | • Identify committed lab members who implement the guidelines (or train those who are open to this) to encourage others<br>• Educate lab members on their role, required changes to their practice, and considerations for implementing the guidelines<br>• Provide supervision to lab members or access institutional services providing consultation (e.g., methods and/or biostatistical services) | • Collaborate with other laboratories or departments (e.g., Animal Care Facilities) to develop systems to assist with elements such as blinding and randomization<br>• Designate processes for review of study protocols and manuscripts to check adherence to the NIH Principles and Guidelines, ARRIVE, PREPARE, etc.<br>• Encourage/establish collaborations and partnerships to share resources |
| **Implementation** | • Share educational information and materials with colleagues<br>• Provide educational outreach on the issue through meetings and presentations to colleagues<br>• Work to educate colleagues of importance on issues and prime for acceptance of required behavioral changes necessary for adoption | • Provide training through a variety of formats to hone skills (e.g., hands on workshops, online modules)<br>• When implementing the guidelines, pilot changes to lab processes and practices to assess for ways to improve<br>• Start with implementing a few practices, rather than all at once and be open to feedback on barriers and enablers to implementation from stakeholders within the lab environment | • Make changes to the lab/provide equipment to assist with blinding, randomization, reporting guidelines, etc.<br>• Phase implementation with pilot projects before moving to system-wide changes<br>• Regularly check-in to assess how implementation is progressing, make changes to strategies if needed on an ongoing basis to balance between lab-specific needs/feasibility and maintaining adherence to guidelines |

Strategies, explanations and elaborations derived, adapted and/or informed by the BCW [23] and the ERIC project terminology and definitions [22].

ARRIVE, Animal Research: Reporting of In Vivo Experiments; BCW, Behaviour Change Wheel; NIH, National Institutes of Health.

implementation strategies focus on providing researchers with a wide array of specific actions that they can implement (e.g., at the investigator and/or trainee level). However, here we also discuss potential other stakeholders who could play a role in implementation of the guidelines (including journals, academic institutions, animal care facilities, and funding agencies).

An important barrier to researchers, related to knowledge, was the variability in awareness and in the current practices of participants. Although a number of participants indicated that they had prior experience or already follow all or most of the guidelines, a number of assessments of preclinical literature have demonstrated low reporting of core practices such as randomization (0% to 47%), blinding (2% to 57%), and sample size calculations (0% to 7%) [32–36]. Indeed, even among our participants, several expressed not currently implementing some of these specific practices. We also identified some misconceptions or areas for improvement for some practices, such as more rigorous approaches to randomization (implementing true randomization versus pseudo-randomization).

Beliefs about consequences, capabilities, and environmental context and resources were also found to influence adoption of the core practices and presented as a barrier to regular and consistent implementation by researchers. Though many participants expressed high intention to implement the guidelines, or that they already implemented some of the guidelines,

feasibility appeared to be a key factor influencing capability to adopt the guidelines. The belief that there are costs and challenges that hinder researchers from implementing the guidelines and practices presents another possible misconception. Indeed, evidence suggests that appropriate experimental design and statistical analysis techniques are key to minimizing the use of animals and reducing waste in research [37,38]. Nonetheless, the feasibility issues associated with the practices are valid concerns as laboratories are already constrained by time and resources. If researchers are already over-burdened with research-related work, they may overestimate the work/risk associated with performing additional tasks. Participants also expressed that the aim of the research may influence implementation of the guidelines (e.g., confirmatory versus exploratory research) or that in some cases certain practices may not be possible (e.g., animal model does not allow for blinding). Initiatives could therefore prioritize implementation of practices in confirmatory research and emphasize that transparent reporting should be paramount, irrespective of whether certain practices have been implemented (e.g., report what was done and explain why a practice could not be implemented). We also note that implementation of certain aspects of the guidelines do not take any resources, but have significant implications on the statistics and analysis of experiments, such as the distinction between biological and technical replicates. We therefore similarly suggest that educational initiatives underscoring this information would be helpful.

Conflicting observations across our study findings (e.g., high motivation and confidence in implementing the guidelines, but numerous noted challenges and need for further resources/ support) may suggest possible overestimation of abilities and resource constraints. Current systems largely reward quantity (e.g., of published papers) over the rigor and reporting. Thus, funders could spearhead more initiatives that highlight and reward implementation of practices to improve rigor and reporting. Ultimately, such rigor and transparency will facilitate proper evaluation by the scientific community. Notably, despite costs, constraints, and variable knowledge, participants were motivated to implement the core practices as they believed that they would lead to positive outcomes and benefits. These outcomes were chiefly concerned with "good scientific practice" and increasing the quality of their work and trust in their findings. This further supports the notion that with education, training, support, and incentives researchers would be likely to implement core practices.

Another overarching theme that has emerged from the findings is the idea that formal training may empower researchers to implement the core practices. To enable this, research institutions and investigators will need to ensure appropriate training, particularly early in an investigator's research career; standardized training may promote the uptake of the core practices (e.g., similar to WHMIS or animal care certification). Resources to support education and training could include access to specialists such as statisticians, information specialists, and methods centers. Incorporating in-house expertise may further support researchers, such as having a designated lab or institutional member with expertise to train new members or check over the study design/manuscript [39,40]. Institutes may also be able to draw from existing resources, including online videos [17] and resources for improving experimental rigor put forth by the NIH [39], educational courses by the Enhancing Quality In Preclinical Data (EQIPD) initiative [18], and the online platform currently in development by our team (www. preclinicaldesign.ca).

Two recent papers assessing perspectives, barriers, and enablers to rigorous reporting in preclinical research (one focused on complete reporting, while the other looked at preregistration of animal studies) came to similar conclusions; various stakeholders involved in preclinical research may be able to encourage implementation of these behaviors by collaborating and creating a system that rewards or incentivizes (e.g., influences hiring, mandates) [41,42]. We believe our finding that researchers are motivated to implement the guidelines, yet continue to

face resource and education-related challenges in doing so, also suggests action by additional stakeholders will be required. Funders and journals could adopt strategies to encourage the implementation of the guidelines and core practices. Encouragingly, there is ample evidence demonstrating that when journals endorse and/or enforce reporting guidelines, the reporting of the studies they publish improves [43]. From this evidence, it is conceivable that planning and reporting of preclinical experiments could be improved if journals were to adopt and enforce harmonized reporting guidelines such as the NIH guidelines [12], ARRIVE 2.0 guidelines [1,31], or PREPARE [44]. ARRIVE 2.0, in particular, has a set of "essential 10" elements that could be emphasized by stakeholders. As examples of these strategies in action, BMBF, a German funding agency, has recently issued a funding competition specifically for preclinical "confirmatory" studies [45], and NIH has also issued several funding competitions focused on improving education in principles of experimental rigor [46]. Institutions may also be able to build on the Declaration on Research Assessment (DORA) initiative [47] by signing and agreeing to base policies that reward quality (over other quantity-based metrics).

A recent editorial by an ethnographer also provided perspectives on implementing methods such as blinding in preclinical research [48]. Main considerations for blinding included "integrity, collegiality, and animal welfare" (e.g., wanting to avoid making mistakes, recruiting others to assist, and disrupting animal care practices). Similar to our findings of feasibility concerns, these considerations point to the need for system-level changes and support and action by additional stakeholders. As such, changing behavior will require a multicomponent strategy. Future research investigating roles, barriers, and enablers faced by other stakeholders (such as those identified within our study and by others [49]) may help to provide a more comprehensive view of the actions needed to enable improved implementation of rigorous experimental study designs and transparent reporting.

## Strengths and limitations

Although our study used a well-established tool for supporting behavior-change initiatives, limitations should be considered. First, as with any interview-based study, our results depend on self-reporting. We acknowledge that our results may be influenced by social desirability bias, wherein responses may lean towards what participants feel the interviewer would like to hear. We aimed to address this by emphasizing at the beginning of the interview that there were no "right" or "wrong" answers and that we were simply interested in interviewee perspectives to help inform the design of resources. Second, our sample largely included cancer and cardiovascular researchers; this presents a potential bias as views expressed may not reflect those of researchers in other biomedical disciplines. We also acknowledge that a limitation of our study is a focus largely on Canadian academic institutions (reflected by sampling through our 2 Canadian funders). Nonetheless, Canadian biomedicine does follow an archetype of academic research that is widely adopted (i.e., in terms of funding, lab organization, institutional support, and incentives that drive academic research). Further research evaluating the feasibility of implementing such behavior change strategies across different institutions, as well as the potential role of other key stakeholders, is warranted. Building on the complex and discursive information elicited from the present study, a future large-scale survey of preclinical researchers across a greater number of research areas could confirm findings and/or identify additional barriers and enablers.

## Conclusion

Through use of our theory-informed approach, we have identified theoretical domain-relevant themes that influence behaviors and potential implementation strategies that can increase the effectiveness of future interventions. Our results suggest that behavior change strategies aimed

at education/persuasion, training/modeling, and environmental restructuring could help preclinical researchers to prepare, build capacity for, and implement rigorous experimental study designs and transparent reporting. This could encourage an overall shift in practice to improve the quality of preclinical research design and reporting, which could in turn facilitate replication and evaluation of studies by others in the research community.

## Materials and methods

### Ethics and reporting

This study was approved through the Ottawa Health Science Network Research Ethics Board (Protocol 20170659-01H) and complies with the Canadian Tri-Council Policy Statement: Ethical Conduct for Research Involving Humans. The study protocol was posted a priori to the Open Science Framework (https://osf.io/dth7b/) and has been reported according to the Consolidated Criteria for Reporting Qualitative Research (COREQ; S2 File) [50].

### Study design

Semi-structured interview guides with open questions and prompts were developed based on guidance for development of TDF-based interviews [21]. Questions aimed to elicit preclinical researchers' views on potential barriers and enablers to application of the 7 core items of rigor and reproducibility promoted by the NIH (definition of the behavior of interest is further elaborated in S3 File). Questions were developed based on TDF version 2 domains [19], as well as the Nature of Behavior domain from TDF version 1 [20]. Interview guides were also reviewed with 2 target stakeholders to improve clarity and wording of the questions and then piloted with 5 preclinical researchers (final interview guides can be found in S4 File).

### Recruitment and procedure

We identified and recruited cancer or cardiovascular researchers through 2 federally supported biomedical research funders (Canadian Vascular Network, BioCanRx) and snowball sampling. Principal investigators, postdoctoral fellows, senior graduate students (i.e., those in the late stages of their doctoral research), and highly qualified personnel (e.g., senior research associates with a PhD) were eligible to participate and approached by email. The total number of participants recruited was informed by the pre-determined 10+3 stopping rule [51]. We aimed to recruit a minimum of 13 participants in 4 groups (cancer researchers, cardiovascular researchers; and principal investigators and other researchers). Participants provided written informed consent and were subsequently interviewed in person or by telephone between May 16, 2018 to January 8, 2019, by a trained research assistant with extensive interview experience and a background in health science and laboratory research (MF). Participants were provided with a handout during the interview, which outlined the NIH Principles and Guidelines for Reporting Preclinical Research (either in person or by email depending on the format of the interview; see S4 File). Interviews were recorded, transcribed verbatim, and identifying information was removed to pseudonymize the transcripts.

### Data analysis

Transcripts were imported to the qualitative software program NVivo 11. We performed directed content analysis [21], wherein the TDF domains were used to organize and codify the data. First, quotes from the interview transcripts were assigned to the relevant TDF domains [52]. Quotes could be assigned to multiple domains if it was felt that more than 1 domain was represented. The first 11 interviews were codified in duplicate by 2 independent researchers

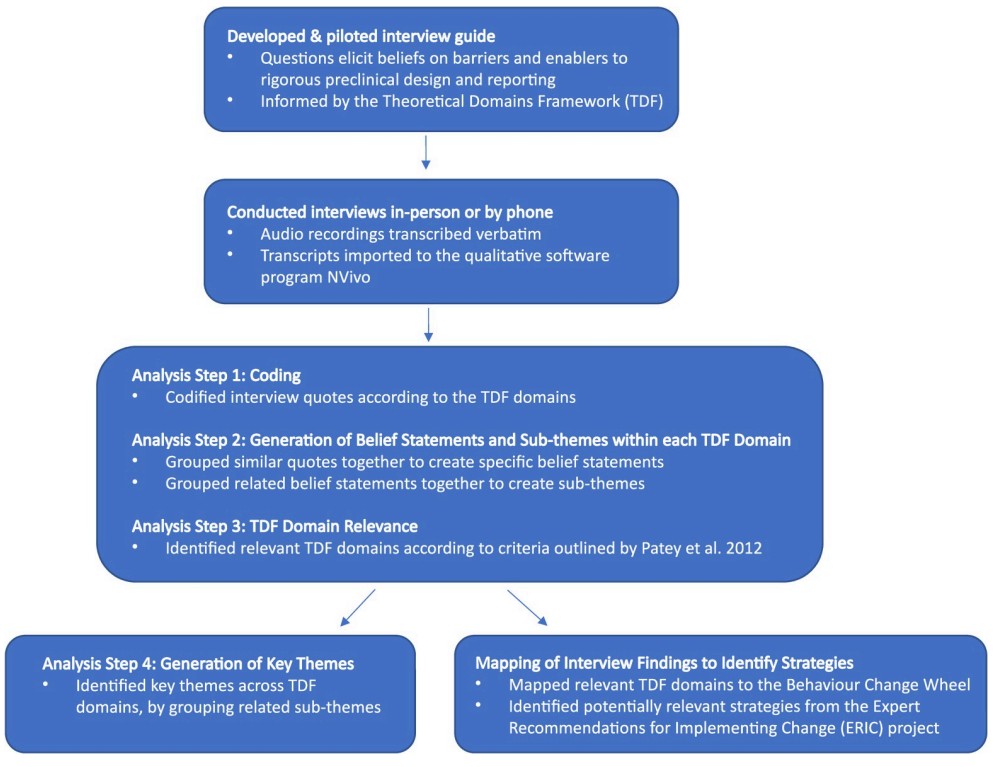

**Fig 2. Study workflow.** ERIC, Expert Recommendations for Implementing Change; TDF, Theoretical Domains Framework.

(MF + AC or VH), and any conflicts were resolved through a consensus discussion. The remaining 19 interviews were then codified by 1 researcher (MF). All coders were trained in application of the TDF to interview studies. In the second stage of analysis, the codified data was reviewed according to each TDF domain. Within each domain, similar quotes were grouped together and described using an overarching "belief statement" (specific belief representing the underlying meaning of the quotes). Similar belief statements were also grouped together to create broader themes. In step 3, the results were reviewed to determine which TDF domains were most relevant to the behavior of interest (i.e., to adopt preclinical guidelines). This analysis was based on frequency of beliefs, strength of beliefs, and conflicting beliefs as outlined by Patey and colleagues [53]. Once relevant domains were identified, global themes were generated to describe high level, interesting features of the data, spanning across the domains. Tables were created to display the identified global themes, sub-themes, belief statements, frequency of these belief statements, and the relevant domains (S5 File). An overview of our study design and analysis approach can be found in Fig 2.

## Identification of potentially relevant implementation strategies

To identify possible implementation strategies, TDF domains from the interview study were mapped to the BCW (S6 File) [21,23]. This framework provides a comprehensive overview of intervention functions and policies, linked to conditions for behavior (capability, opportunity, and motivation) [23]. We then identified potentially relevant implementation strategies from the ERIC project [22]. These were coded by 1 researcher (MF) and audited by 2 independent team members (MSJ or AS). Conflicts were resolved through consensus with all 3 researchers, and consultation with a fourth reviewer (MML) as needed.

## Supporting information

**S1 File. Description of Theoretical Domains Framework domains.**
(PDF)

**S2 File. Consolidated Criteria for Reporting Qualitative Studies (COREQ) checklist.**
(PDF)

**S3 File. Definition of behavior using the Action, Actor, Context, Target, Time (AACTT) framework.**
(PDF)

**S4 File. Interview guide for researchers.**
(PDF)

**S5 File. Sub-themes, belief statements and frequencies.**
(PDF)

**S6 File. Detailed description of mapping exercise methods and results.**
(PDF)

**S7 File. Mapping of TDF domains to the Behaviour Change Wheel Intervention Functions and Policies.**
(PDF)

**S8 File. Behaviour Change Wheel and Expert Recommendations for Implementing Change (ERIC) Mapping (Final Consensus).**
(PDF)

**S9 File. Organization and categorization of Expert Recommendations for Implementing Change (ERIC) implementation strategies.**
(PDF)

**S1 Data. Supplementary data.**
(XLSX)

## Author Contributions

**Conceptualization:** Manoj M. Lalu, Justin Presseau, Dean A. Fergusson.

**Formal analysis:** Madison K. Foster.

**Funding acquisition:** Manoj M. Lalu, Justin Presseau, Kelly D. Cobey, Jamie C. Brehaut, Dean A. Fergusson.

**Investigation:** Madison K. Foster, Victoria T. Hunniford, Analyssa Cardenas.

**Methodology:** Manoj M. Lalu, Justin Presseau, Madison K. Foster, Jamie C. Brehaut, Dean A. Fergusson.

**Project administration:** Joshua Montroy.

**Resources:** Manoj M. Lalu, Dean A. Fergusson.

**Supervision:** Manoj M. Lalu, Dean A. Fergusson.

**Validation:** Ayni Sharif, Matthew S. Jeffers.

**Visualization:** Matthew S. Jeffers.

**Writing – original draft:** Manoj M. Lalu, Madison K. Foster, Victoria T. Hunniford.

**Writing – review & editing:** Manoj M. Lalu, Justin Presseau, Madison K. Foster, Kelly D. Cobey, Jamie C. Brehaut, Carolina Ilkow, Joshua Montroy, Analyssa Cardenas, Ayni Sharif, Matthew S. Jeffers, Dean A. Fergusson.

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
