## [Editor Report · Decision Letter 0]

28 Jun 2022

Dear Manoj, 

Thank you for submitting your manuscript entitled "Identifying barriers and enablers to rigorous conduct and reporting of preclinical studies" for consideration as a Meta-Research Article by PLOS Biology.

Your manuscript has now been evaluated by the PLOS Biology editorial staff, as well as by an academic editor with relevant expertise, and I'm writing to let you know that we would like to send your submission out for external peer review.

Once your full submission is complete, your paper will undergo a series of checks in preparation for peer review. After your manuscript has passed the checks it will be sent out for review. To provide the metadata for your submission, please Login to Editorial Manager (https://www.editorialmanager.com/pbiology) within two working days, i.e. by Jun 30 2022 11:59PM.

Kind regards,

Roli

Roland Roberts, PhD

Senior Editor

PLOS Biology

rroberts@plos.org

---

## [Decision Letter · Decision Letter 1]

5 Aug 2022

Dear Manoj,

Thank you for your patience while your manuscript "Identifying barriers and enablers to rigorous conduct and reporting of preclinical studies" was peer-reviewed at PLOS Biology. It has now been evaluated by the PLOS Biology editors, an Academic Editor with relevant expertise, and by three independent reviewers. 

You’ll see that the three reviewers are broadly positive about the study, but they each raise a number of concerns that must be addressed. Some common themes emerge, including the potential for more incisive analysis (reviewers #1 and #2), and the need for more information about interviewer diversity (all three reviewers). The Academic Editor recognises that you may not be able to comply with reviewer #2's request about raw data availability, depending on whether this is compatible with your confidentiality and data protection rules.

In light of the reviews, which you will find at the end of this email, we would like to invite you to revise the work to thoroughly address the reviewers' reports.

Given the extent of revision needed, we cannot make a decision about publication until we have seen the revised manuscript and your response to the reviewers' comments. Your revised manuscript is likely to be sent for further evaluation by all or a subset of the reviewers.

**IMPORTANT - SUBMITTING YOUR REVISION**

*Re-submission Checklist*

*Published Peer Review*

*PLOS Data Policy*

Sincerely,

Roli

Roland Roberts, PhD

Senior Editor

PLOS Biology

rroberts@plos.org

REVIEWERS' COMMENTS:

Reviewer #1:

Summary of research

"Preclinical guidelines" such as ARRIVE inform researchers conducting (animal) experiments and readers of (their) scholarly publications on matters of scientific rigour. They further enable reviewers and readers to adequately assess and evaluate the methodological robustness of the research study presented to ensure its reproducibility. 

Referring to the poor adoption of such guidelines by preclinical researchers, the study presented by Manoj et al. aims to 

A) identify factors influencing the application of "preclinical guidelines" and 

B) derive implementation strategies, measures and policies for research-funding organizations and/or research-performing organizations to improve reproducibility in preclinical research. 

Hence, the manuscript's subject matter is relevant to the research community and target groups PLOS Biology addresses. 

Manoj et al. chose a qualitative research design interviewing thirty expert researchers working in cancer and cardiovascular research. To identify factors that influence the adoption of research practices (standards, replication, statistics, randomization, blinding, sample estimation, inclusion and exclusion criteria derived from NIH principles and guidelines for reporting preclinical research) supposed to improve the reproducibility of preclinical research, the authors addressed 14 behavioural domains synthesized from 128 constructs derived from 33 theories of behaviour and behaviour change clustered the 'Theoretical Domain Framework', TDF, [Ref. 43] using a semi-structured interview guide. 

Manoj et al. used a deductive approach and coded interviewees' quotes based on TDF domains but determined their relevance through quantitative, here frequency analysis. Quotes were further aggregated in overarching "belief statements" representing their meaning and clustered in one "enabler theme" and two "barrier themes". (A) 

To identify appropriate implementation strategies, they mapped TDF domains identified as relevant to the "Behavioral Change Weel" [Ref 21, 29] and the "Expert Recommendation for Implementing Change" project [Ref 28], with the latter referred to as two tools for selecting/designing implementation strategies that lead to behavioural change in individuals. (B)

Briefly, behavioural change requires that individuals become aware of a problem that gives them a 'reason' to reconsider a particular behaviour. The individual may weigh the costs and benefits of alternative behaviours and form an attitude (consisting of cognitive, affective, and normative evaluative elements) that leads to the intention to make a decision, either to change or to maintain a particular behaviour. Importantly, behavioural change usually occurs in a social setting (team, department, organization) and is supposed to be followed by various social and contextual responses that may or may not reinforce the behaviour. Last but not least, external factors such as organizational/national/supra-national policies or funding mechanisms influence the interviewees' current research practices addressed in the study.

Major issues with regard to A)

To get deeper insights and understand the interviewees' current behaviour /behavioural intentions intention, the authors may consider presenting case summaries based on a code-document/case-matrix and performing/presenting a qualitative case comparison analysis. Further, Manoj et al. could apply a more data-driven approach, i.e. inductive development of categories and codes. 

Major issues with regard to B)

Considering the study's limited sample size, Manoj et al. would need to provide information on how many research-performing organizations the interviewees represent and whether preferentially Canadian or international organizations were chosen. With respect to the complexity of (social) behaviour, the authors would need to provide a plausible argumentation of their generalization claims concerning the identification of organizational implementation strategies. 

Other major issues

To strengthen the transparency of the study and ensure the reproducibility of the authors' analysis, it would be helpful for readers and reasonable for researchers who may want to re-analyze or re-use the study's data if Manoj et al. share their data. Among other things, this includes anonymized interview transcripts (approved by interviewees), the study's final category system (coding frame), a code-document matrix, and a detailed analysis protocol.

Reviewer #2:

There is a real need to explore how to drive change to implement robust experimental design as standard in the in vivo space. This manuscript looks to explore the blockers through qualitative research and provide a framework to move the community forward. As a scientist working actively in this area, this manuscript should have high interest to me. However, it didn't quite land and the analysis of the interviews seemed superficial and uninformative. The paper was also not cohesive and this could be domain specialist not articulating clearly enough to researcher from other fields. There are also some sections that could perpetuation misunderstandings and hinders the implementation of robust experimental design. 

Major: 

1. This article will be read by scientist who are not qualitative research specialist. It needs to be assessable to this audience. Review the text to ensure accessibility for the target audience. Please also include a schematic of the workflow to give a big picture explanation of the work presented in this article

e.g. Survey construct focused on TDF domains - survey - coding - summation of observations - Behavioural wheel. 

2. The paper for a non expert isn't cohesive. For example

A: In supplementary file 1: there are two separate tables both describing domains but in different ways. One has a series of definitions and the other is a series of questions. I don't understand how these were used in the analysis. I original though that the first set were used to classify the response provided to the survey question. I then don't understand how nature of the behaviour fits into this process.

B: Supplement File 4: We now have construct statement associated with each section. Sometimes this overlaps with the TDF domains but sometimes doesn't. How does this fit in? 

3. Explorative versus confirmation experiments. 

This is a hazardous area. There is not consensus on the definitions. For some communities, if you class the work as explorative then you should not apply any statistical tests. In my experience, majority of research is explorative and not confirmation (where you are looking to replicate a known ES). The current wording states that if you are in explorative mode you cannot do a power calculation. That is not true. Consider an omic experiment using ex-vivo samples, we still conduct power analysis we input a SD metric that encompasses 75% of the variables of interest and define an effect size of interest. Alternatively, where you have multiple outcome variables, you can talk about power in terms of cohen's d and reflect on predicted ES for this intervention. The paper is contributing to the misunderstanding around power calculations. 

3. Inclusion and exclusion criteria.

The definition includes both the criteria around animals and data. On page 8, a quote is included that says it isn't relevant because they are in bred mice. This is focused only on one aspect of the criteria - the animal not the data. The presentation in this section is at real risk of perpetuating the misunderstanding of the definition and intention around this criteria. 

4: Superficial reflection

Examples

a: The survey seems very interest and explores many different drivers of behaviour e.g. self efficacy etc but this doesn't then come through in the evaluation. If the survey results were framed in the language of construct or TDF - this would add more reflection on the drivers behind this behaviour. Just is saying what we already know. Adding more reflection on the drivers would have been interesting.

b: Randomisation. During the interview the researchers said they were confident to roll out these methods and very often had experience of doing it. However, the paper highlights that the randomisation implemented wasn't true randomisation. The missing point - is that the scientists are over stating their abilities - probably due to a lack of knowledge. This means the self reporting is also a weakness of this analysis. 

c. The scientists stated they were confident and yet most do not routinely make the distinct ion between biological and technical replicates. This process doesn't have direct practical costs but significant statistical implications. If they are confident they should do it, and they believe in the value then they should do it. If they are not, then it probably due to lack of analysis skills etc.

d. We know a lot of these issues - none of this is a surprise. To drive change we need hard number. E.g. "multiple participants shared they had not received formal training" what proportion? What proportion had not heard of any guidelines? 

e. "Most stated they were very confident or confident in their capabilities to apply all the experimental design methodology" - OK if that was true then it is at odds with a lot of other details that are presented. For example asking for training, asking for statisticians etc. This paper has no reflection on these conflicting observations. Is this arising from peoples first reaction to reflecting on change is to throw out reasons why it can't be done and it impossible and the process of you asking is leading to them feeling actually they could do it and really should. The quotes suggests your interview took people on that journey. How does that help us think about the next steps? 

4: This paper is structured around the NIH guidelines. Why then does the abstract reference the ARRIVE guidelines?

5: Limitations - I assume all these researchers are academic? US based? This isn't presented in table 1 and needs discussing in the limitations section. 

Minor

1. Supplementary File 1: behavioural regulation - the definition was not accessible to me. 

2. When talking about guidelines what about PREPARE? 

3. ARRIVE 2.0 has a essential set for journals to prioritise - this should mentioned in the section where this is discussed. 

4. Table 2 - "Look for and apply to funding opportunities prioritizing practices...." I found it hard to parse and wasn't sure what you meant. Do you mean apply to funding bodies that require you to follow good practice as default?

5. Quite a few sentences are hard to parse eg

a. "As we only conducted interviews with preclinical researchers … " page 15. 

b. "Nonetheless, the feasibility issues associated with the practices discussed by …" page 16. 

6. Data availability - I would dispute that all data has been made available as the transcripts have not. Please reword.

Reviewer #3:

[identifies himself as Timothy M. Errington]

The manuscript by Lalu et al. describes results from multiple structured interviews from preclinical researchers (specifically cancer and cardiovascular researchers who perform in vivo research) on the topic of implementing and reporting preclinical research guidelines. They identified four global theme from these interviews (two barriers and two enablers) that identify opportunities to inform strategies and approaches in improving the conduct and reporting of preclinical research.

Overall, I thought this was a well written manuscript on an important and timely topic that will be of interest to a broad audience. I enjoyed reading it and the excerpts from the interviews give great context to what the authors found in their study. Below are suggests for the authors to consider in improving their manuscript:

[1] The authors recruited from Canadian funders, does that mean the researchers who participated were also from Canadian institutions or were they more geographically distributed? Similarly, how broad were the participants in terms of institutions (e.g., were many at the same institution or were all from different institutions)? The reason is how broadly applicable these points of view are. As the authors indicated a limitation of the current study is that it only included cancer and cardiovascular researchers, but it is likely more limited than that - such as only academic researchers (e.g., were industry researchers invited) at Canadian institutions. Any additional information on this could be included in the methods, limitations section, and table 1 where applicable.

[2] On page 9 a common concern was expressed by participants that the rigor elements in the reporting guidelines (e.g., increased sample sizes) would lead to less experiments, which would be perceived as a decrease in productivity (e.g., less papers). This tension (e.g., many lower quality experiments vs less higher quality experiments) is a central factor not currently discussed in the paper. The discussion includes how funders and journals could adopt strategies to encourage implementation of guidelines and core practices, but not how to shift this incentive concern being raised that is a major barrier to adoption.

Minor points:

[1] In the abstract, listing the themes (e.g., Barrier Theme 1) seems unnecessary as the context is missing. I think the authors could delete these as the narrative is sufficient for the abstract with the theme numbers more relevant for the body of the paper. Instead the authors could state that there were four global themes identified, two that reflect barriers and two that reflect enablers.

[2] The authors might consider these papers that also looked at barriers and enablers to conducting preclinical animal research: https://osf.io/wvn85/ & https://journals.plos.org/plosone/article?id=10.1371/journal.pone.0226443

[3] The authors might consider this paper on supporting rigor champions (https://elifesciences.org/articles/55915).

[4] Related to the above paper and the topic in the paper of enabling rigor champions within the lab, the authors might consider the relevance of this papers' approach to shifting the focus from educating individuals to changing the culture within the lab: https://www.pnas.org/doi/10.1073/pnas.1917848117

[5] The authors posted their protocol file to OSF on July 11, 2018. The file says the start date was May 8, 2018 and completion October 21, 2018 - so does this mean the study protocol was posted half-way through the data collection? As far as I can tell the authors didn't preregister (e.g., it is not available here: https://osf.io/registries), so I am going off of log/history of the file. This also suggests the interview data come from 2018 - the authors might consider adding what, if anything, has changed from when these interviews were performed vs now. Also, I suggest indicating 2018 was when the interviews were performed in the papers methods section for increased transparency for the reader.

[6] Related to the point above, in methods the authors state their sampling strategy was pre-determined, yet there is no indication of the approach in the OSF file.

---

## [Editor Report · Decision Letter 2]

14 Nov 2022

Dear Manoj,

Thank you for your patience while we considered your revised manuscript "Identifying barriers and enablers to rigorous conduct and reporting of preclinical studies" for publication as a Meta-Research Article at PLOS Biology. This revised version of your manuscript has been evaluated by the PLOS Biology editors, and the Academic Editor.

Based on our Academic Editor's assessment of your revision, we are likely to accept this manuscript for publication, provided you satisfactorily address the following data and other policy-related requests.

IMPORTANT: Please address the following:

a) Competing interests - you currently list the following competing interest: "MML has previously acted as a guest editor for PLOS Biology." We would not normally consider this as a competing interest, but are fine if you want to leave it as-is.

b) Ethics - You say that the study "complies with all relevant ethical regulations." but do not specify which national/international guidelines the study adheres to (e.g. Declaration of Helsinki) - please could you clarify this in the text?

c) Data availability - we agree that we should not insist on you providing the full transcripts, as requested by R1, for the reasons that you mention in your rebuttal. However, Table 1 has some values for which you only present median and range values (age of participants, experience, length of interview); please could you supply the underlying numerical values for these as a supplementary data file, S1_Data.xslx?

d) Please cite the location of the data clearly in the legend to Table 1, e.g. “The data underlying this Figure can be found in S1 Data”.

We expect to receive your revised manuscript within two weeks. 

*Published Peer Review History*

*Press*

Sincerely,

Roli

Roland Roberts, PhD

Senior Editor,

rroberts@plos.org,

PLOS Biology

DATA NOT SHOWN?

---

## [Editor Report · Decision Letter 3]

24 Nov 2022

Dear Manoj,

Thank you for the submission of your revised Meta-Research Article "Identifying barriers and enablers to rigorous conduct and reporting of preclinical studies" for publication in PLOS Biology. On behalf of my colleagues and the Academic Editor, Ulrich Dirnagl, I'm pleased to say that we can in principle accept your manuscript for publication, provided you address any remaining formatting and reporting issues. These will be detailed in an email you should receive within 2-3 business days from our colleagues in the journal operations team; no action is required from you until then. Please note that we will not be able to formally accept your manuscript and schedule it for publication until you have completed any requested changes.

Sincerely, 

Roli

Senior Editor

PLOS Biology

rroberts@plos.org